# Unsupervised Learning of Structured Representations via Closed-Loop Transcription

## Abstract

This paper proposes an unsupervised method for learning a unified representation that serves both discriminative and generative purposes. While most existing unsupervised learning approaches focus on a representation for only one of these two goals, we show that a unified representation can enjoy the mutual benefits of having both. Such a representation is attainable by generalizing the recently proposed *closed-loop transcription* framework, known as CTRL, to the unsupervised setting. This entails solving a constrained maximin game over a rate reduction objective that expands features of all samples while compressing features of augmentations of each sample. Through this process, we see discriminative low-dimensional structures emerge in the resulting representations. Under comparable experimental conditions and network complexities, we demonstrate that these structured representations enable classification performance close to state-of-the-art unsupervised discriminative representations, and conditionally generated image quality significantly higher than that of state-of-the-art unsupervised generative models.

## 1 Introduction

In the past decade, we have witnessed an explosive development in the practice of machine learning, particularly with deep learning methods. A key driver of success in practical applications has been marvelous engineering endeavors, often focused on fitting increasingly large deep networks to input data paired with task-specific sets of labels. Brute-force approaches of this nature, however, exert tremendous demands on hand-labeled data for supervision and computational resources for training and inference. As a result, an increasing amount of attention has been directed toward using self-supervised or unsupervised techniques to learn representations that can not only learn without human annotation effort, but also be shared across downstream tasks.

**Discriminative versus Generative.** Tasks in unsupervised learning are typically separated into two categories. Discriminative ones frame high-dimensional observations as inputs, from which low-dimensional class or latent information can be extracted, while generative ones frame observations as generated outputs, which should often be sampled given some semantically meaningful conditioning.

Unsupervised learning approaches targeted at discriminative tasks are mainly based on a key idea: to pull different views from the same instance closer while enforcing a non-collapsed representation by either contrastive learning techniques (Chen et al., 2020b; He et al., 2020; Grill et al., 2020a), covariance regularization methods (Bardes et al., 2021; Zbontar et al., 2021), or using architecture design (Chen & He, 2020; Grill et al., 2020b). Their success is typically measured by the accuracy of a simple classifier (say a shallow network) trained on the representations that they produce, which have progressively improved over the years. Representations learned from these approaches, however, do not emphasize much about the intrinsic structure of the data distribution, and have not demonstrated success for generative purposes.

In parallel, generative methods like GANs (Goodfellow et al., 2014) and VAEs (Kingma & Welling, 2013) have also been explored for unsupervised learning. Although generative methods have made striking progress in the quality of the sampled or autoencoded data, when compared to the aforementioned discriminative methods, representations learned with these approaches demonstrate inferior performance in classification.

**Toward A Unified Representation?** The disparity between discriminative and generative approaches in unsupervised learning, contrasted against the fundamental goal of learning representations that are useful across many tasks, leads to a natural question that we investigate in this paper: *in the unsupervised setting, is it possible to learn a unified representation that is effective for both*

*discriminative and generative purposes? Further, do they mutually benefit each other?* Concretely, we aim to learn a *structured representation* with the following two properties:

1. The learned representation should be discriminative, such that simple classifiers applied to learned features yield high classification accuracy.

2. The learned representation should be generative, with enough diversity to recover raw inputs, and structure that can be exploited for sampling and generating new images.

The fact that human visual memory serves both discriminative tasks (for example, detection and recognition) and generative or predictive tasks (for example, via replay) (Keller & Mrsic-Flogel, 2018; Josselyn & Tonegawa, 2020; Ven et al., 2020) indicates that this goal is achievable. Beyond being possible, these properties are also highly practical – successfully completing generative tasks like unsupervised conditional image generation (Hwang et al., 2021), for example, inherently requires that learned features for different classes be both structured for sampling and discriminative for conditioning. On the other hand, the generative property can serve as a natural regularization to avoid representation collapse.

**Closed-Loop Transcription via a Constrained Maximin Game.** The class of linear discriminative representations (LDRs) has recently been proposed for learning diverse and discriminative features for multi-class (visual) data, via optimization of the *rate reduction* objective (Chan et al., 2022). In the supervised setting, these representations have been shown to be be both discriminative and generative if learned in *a closed-loop transcription* framework via a maximin game over the rate reduction utility between an encoder and a decoder (Dai et al., 2022). Beyond the standard joint learning setting, where all classes are sampled uniformly throughout training, the closed-loop framework has also been successfully adapted to the *incremental setting* (Tong et al., 2022), where the optimal multi-class LDR is learned one class at a time. In the incremental (supervised) learning setting, one solves a *constrained maximin problem* over the rate reduction utility which keeps learned memory of old tasks intact (as constraints) while learning new tasks. It has been shown that this new framework can effectively alleviate the catastrophic forgetting suffered by most supervised learning methods.

**Contributions.** In this work, we show that the closed-loop transcription framework proposed for learning LDRs in the supervised setting (Chan et al., 2022) can be adapted to a purely unsupervised setting. In the unsupervised setting, we only have to view each sample and its augmentations as a "new class" while using the rate reduction objective to ensure that learned features are both *invariant* to augmentation and *self-consistent* in generation; this leads to a constrained maximin game that is similar to the one explored for incremental learning (Tong et al., 2022). Our overall approach is illustrated in Figure 1.

As we experimentally demonstrate in Section 4, our formulation benefits from the mutual benefits of both discriminative and generative properties. It largely bridges the gap between two formerly distinct set of methods: by standard metrics and under comparable experimental conditions, it enables classification performance close to discriminative methods and unsupervised conditional generative quality significantly higher than state-of-the-art techniques. Coupled with evidence from prior work, this suggests that the closed-loop transcription through the (constrained) maximin game between the encoder and decoder has the potential to offer a unifying framework for both discriminative and generative representation learning, across supervised, incremental, and unsupervised settings.

| Method | Linear Probe | Image Generation | UCIG |
|---|---|---|---|
| SimCLR (Chen et al., 2020b) | ✔ | ✗ | ✗ |
| MOCO-V2 (He et al., 2020) | ✔ | ✗ | ✗ |
| ContraD (Jeong & Shin, 2021) | ✔ | ✔ | ✗ |
| PATCH-VAE (Parmar et al., 2021) | ✔ | ✔ | ✗ |
| CTRL-Binary (Dai et al., 2022) | ✔ | ✔ | ✗ |
| SLOGAN (Hwang et al., 2021) | ✗ | ✔ | ✔ |
| U-CTRL (ours) | ✔ | ✔ | ✔ |

Table 1: Comparison of the downstream task capabilities of different unsupervised learning methods. UCIG refers to Unsupervised Conditional Image Generation (Hwang et al., 2021).

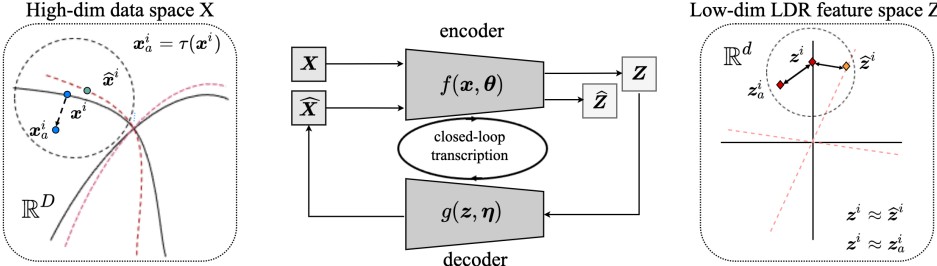

Figure 1: **Overall framework** of closed-loop transcription for unsupervised learning. Two additional constraints are imposed on the Binary-CTRL method proposed in prior work (Dai et al., 2022): 1) self-consistency for sample-wise features $z^i$ and $\hat{z}^i$, say $z^i \approx \hat{z}^i$; and 2) invariance/similarity among features of augmented samples $z^i$ and $z_a^i$, say $z^i \approx z_a^i = f(\tau(x^i), \theta)$, where $x_a^i = \tau(x^i)$ is an augmentation of sample $x^i$ via some transformation $\tau(\cdot)$.

## 2 RELATED WORK

Our work is mostly related to three categories of unsupervised learning methods: (1) self-supervised learning via discriminative models, (2) self-supervised learning via generative models, and (3) unsupervised conditional image generation. Table 1 compares the capabilities of models learned by various representative unsupervised learning methods.

**Self-Supervised Learning for Discriminative Models.** On the discriminative side, works like SimCLR (Chen et al., 2020b), MoCo (He et al., 2020), and BYOL (Grill et al., 2020a) have recently shown overwhelming effectiveness in learning discriminative representations of data. MoCo (He et al., 2020) and SimCLR (Chen et al., 2020b) seek to learn features by pulling together features of augmented versions of the same sample while pushing apart features of all other samples, while BYOL (Grill et al., 2020a) trains a student network to predict the representation of a teacher network in a contrastive setting. BarlowTwins (Zbontar et al., 2021) and TCR (Li et al., 2022) learn by regularizing the covariance matrix of the embedding. However, features learned by this class of methods are typically highly compressed, and *not* designed to be used for generative purposes.

**Self-Supervised Learning with Generative Models.** On the generative side, the original GAN (Goodfellow et al., 2014) can be viewed as a natural self-supervised learning task. With an additional linear probe, works like DCGAN (Radford et al., 2015) have shown that features in the discriminator can be used for discriminative tasks. To further enhance the features, extensions like BiGAN (Donahue et al., 2016) and ALI (Dumoulin et al., 2016) introduce a third network into the GAN framework, aimed at learning an inverse mapping for the generator, which when coupled with labeled images can be used to study and supervise semantics in learned representations. Other works like SSGAN (Chen et al., 2019), SSGAN-LA (Hou et al., 2021), and ContraD (Jeong & Shin, 2021) propose to put augmentation tasks into GAN training to facilitate representation learning. Outside of GANs, variational autoencoders (VAEs) have been adapted to generate more semantically meaningful representations by trading off latent channel capacity and independence constraints with reconstruction accuracy (Higgins et al., 2016), an idea that has also been incorporated into recognition improvements using patch-level bottlenecks (Gupta et al., 2020), which encourage a VAE to focus on useful patterns in images. By incorporating data-augmentation, VAE is also shown to achieve fair discriminative performance (Falcon et al., 2021). Recently, works like MAE (He et al., 2021) and CAE (Chen et al., 2022) have learned representations by solving masked reconstruction tasks using vision transformers. Autogressive approaches like iGPT (Chen et al., 2020a) have also demonstrated decent self-supervised learning performance, which improves further with the incorporation of contrastive learning (Kim et al., 2021). However, unless supervised, features learned by those previously mentioned methods either do not have strong discriminative performance, or cannot be directly exploited to condition the generative task.

**Unsupervised Conditional Image Generation (UCIG).** For generative models, we often want to be able to generate images conditioned on a certain class or style, even in a completely unsupervised setting. This requires that the learned representations have structures that correspond to the desired conditioning. InfoGAN (Chen et al., 2016) proposes to learn interpretable representations by maximizing the mutual information between the observation and a subset of the latent code. ClusterGAN (Mukherjee et al., 2019) assumes a discrete Gaussian prior where discrete variables are defined as a

one-hot vector and continuous variables are sampled from Gaussian distribution. Self-Conditioned GAN (Liu et al., 2020) uses clustering of discriminative features as labels to train. SLOGAN (Hwang et al., 2021) proposes a new conditional contrastive loss (U2C) to learn latent distribution of the data. Note that compared to our work, ClusterGAN and SLOGAN introduce an additional encoder that leads to increased computational complexity. On the VAE side, works like VaDE (Jiang et al., 2016) cluster based on the learned feature of a supervised ResNet. Variational Cluster (Prasad et al., 2020) simultaneously learns a prior that captures the latent distribution of the images and a posterior to help discriminate between data points in an end-to-end unsupervised setting. In this work, we will see how clusters can be estimated in a principled way in a more unified framework, by optimizing the same type of objective function that we use for learning features.

## 3 METHOD

### 3.1 PRELIMINARIES: RATE REDUCTION AND CLOSED-LOOP TRANSCRIPTION

**Assumptions on Data.** Our work, as well as prior work in closed-loop transcription (Dai et al., 2022; Tong et al., 2022), considers a set of $N$ images $\boldsymbol{X} = [\boldsymbol{x}^1, \boldsymbol{x}^2, ..., \boldsymbol{x}^N] \subset \mathbb{R}^D$ sampled from $k$ classes. Borrowing notation from (Yu et al., 2020), the membership of the $N$ samples in the $k$ classes is denoted using $k$ diagonal matrices: $\boldsymbol{\Pi} = \{\boldsymbol{\Pi}_j \in \mathbb{R}^{N \times N}\}_{j=1}^k$, where the diagonal entry $\boldsymbol{\Pi}_j(i, i)$ of $\boldsymbol{\Pi}_j$ is the probability of sample $i$ belonging to subset $j$. Let $\Omega \doteq \{\boldsymbol{\Pi} \mid \sum \boldsymbol{\Pi}_j = \boldsymbol{I}, \boldsymbol{\Pi}_j \geq \boldsymbol{0}.\}$ be the set of all such matrices. WLOG, we may assume that classes are separable, with images for each belonging to a low-dimensional submanifold in the space $\mathbb{R}^D$.

**Unsupervised Discriminative Autoencoding.** The goal of transcription is to learn a unified representation, with the structure required to both classify and generate images from these $k$ classes. Concretely, this is achieved by learning two continuous mappings: (1) an encoder parametrized by $\theta$: $f(\cdot, \theta) : \boldsymbol{x} \mapsto \boldsymbol{z} \in \mathbb{R}^d$ with $d \ll D$ such that all samples are mapped to their features as $\boldsymbol{X} \xrightarrow{f(\boldsymbol{x}, \theta)} \boldsymbol{Z}$ with $\boldsymbol{Z} = [\boldsymbol{z}^1, \boldsymbol{z}^2, ..., \boldsymbol{z}^N] \subset \mathbb{R}^d$, and (2) an inverse map $g(\cdot, \eta) : \boldsymbol{z} \mapsto \hat{\boldsymbol{x}} \in \mathbb{R}^D$ such that $\boldsymbol{x}$ and $\hat{\boldsymbol{x}} = g(f(\boldsymbol{x}))$ is close. In other words, $\boldsymbol{X} \xrightarrow{f(\boldsymbol{x}, \theta)} \boldsymbol{Z} \xrightarrow{g(\boldsymbol{z}, \eta)} \hat{\boldsymbol{X}}$ forms an autoencoding.

In this work, we specifically learn this mapping in an *entirely unsupervised* fashion, without knowing the ground-truth class labels $\boldsymbol{\Pi}$ at all. As stated in the introduction, a both discriminative and generative representation is difficult to achieve by standard generative methods like VAEs and GANs. This is one of the motivations for the closed-loop transcription framework (CTRL) proposed by (Dai et al., 2022), which we will generalize to the unsupervised setting.

**Maximizing Rate Reduction.** The CTRL framework (Dai et al., 2022) was proposed for the supervised setting, where it aims to map each class onto an independent linear subspace. As shown in (Yu et al., 2020), such a linear discriminative representation (LDR) can be achieved by maximizing a coding rate reduction objective, known as *the MCR$^2$ principle*:

$$\Delta R(\boldsymbol{Z} | \boldsymbol{\Pi}) \doteq \underbrace{\frac{1}{2} \log \det \left( \boldsymbol{I} + \frac{d}{N \epsilon^2} \boldsymbol{Z} \boldsymbol{Z}^\top \right)}_{R(\boldsymbol{Z})} - \underbrace{\sum_{j=1}^k \frac{\mathsf{tr}(\boldsymbol{\Pi}_j)}{2N} \log \det \left( \boldsymbol{I} + \frac{d}{\mathsf{tr}(\boldsymbol{\Pi}_j) \epsilon^2} \boldsymbol{Z} \boldsymbol{\Pi}_j \boldsymbol{Z}^\top \right)}_{R^c}. \quad (1)$$

where each $\boldsymbol{\Pi}_j$ encodes the membership of the $N$ samples described before. As discussed in (Chan et al., 2022), the first term $R(\boldsymbol{Z})$ measures the total rate (volume) of all features whereas the second term $R^c$ measures the average rate (volume) of the $k$ components. Our work adapts this formula to design meaningful objectives in the unsupervised setting.

**Closed-Loop Transcription.** To learn the autoencoding $\boldsymbol{X} \xrightarrow{f(\boldsymbol{x}, \theta)} \boldsymbol{Z} \xrightarrow{g(\boldsymbol{z}, \eta)} \hat{\boldsymbol{X}}$, a fundamental question is how we measure the difference between $\boldsymbol{X}$ and the regenerated $\hat{\boldsymbol{X}} = g(f(\boldsymbol{X}))$. It is typically very difficult to put a proper distance measure in the image space (Wang et al., 2004). To bypass this difficulty, the *closed-loop transcription* framework (Dai et al., 2022) proposes to measure the difference between $\boldsymbol{X}$ and $\hat{\boldsymbol{X}}$ through the difference between their features $\boldsymbol{Z}$ and $\hat{\boldsymbol{Z}}$ mapped through the same encoder:

$$\boldsymbol{X} \xrightarrow{f(\boldsymbol{x}, \theta)} \boldsymbol{Z} \xrightarrow{g(\boldsymbol{z}, \eta)} \hat{\boldsymbol{X}} \xrightarrow{f(\boldsymbol{x}, \theta)} \hat{\boldsymbol{Z}}. \quad (2)$$

The difference can be measured by the rate reduction between $\boldsymbol{Z}$ and $\hat{\boldsymbol{Z}}$, a special case of (1) with $k = 2$ classes:

$$\Delta R(\mathbf{Z}, \hat{\mathbf{Z}}) \doteq R(\mathbf{Z} \cup \hat{\mathbf{Z}}) - \frac{1}{2}\big(R(\mathbf{Z}) + R(\hat{\mathbf{Z}})\big). \tag{3}$$

Such a $\Delta R$ is a principled distance between subspace-like Gaussian ensembles, with the property that $\Delta R(\mathbf{Z}, \hat{\mathbf{Z}}) = 0$ iff $\mathrm{Cov}(\mathbf{Z}) = \mathrm{Cov}(\hat{\mathbf{Z}})$ (Ma et al., 2007).

As shown in (Dai et al., 2022), applying this measure in the closed-loop CTRL formulation can already learn a decent autoencoding, even without class information. This is known as the CTRL-Binary program:

$$\max_{\theta} \min_{\eta} \quad \Delta R(\mathbf{Z}, \hat{\mathbf{Z}}) \tag{4}$$

However, note that (4) is practically limited because it only aligns the dataset $\mathbf{X}$ and the regenerated $\hat{\mathbf{X}}$ at the distribution level. There is no guarantee that for each sample $\mathbf{x}$ would be close to the decoded $\hat{\mathbf{x}} = g(f(\mathbf{x}))$. For example, (Dai et al., 2022) shows that a car sample can be decoded into a horse; the so obtained (autoencoding) representations are not sample-wise *self-consistent*!

### 3.2 Sample-Wise Constraints for Unsupervised Transcription

To improve discriminative and generative properties of representations learned in the unsupervised setting, we propose two additional mechanisms for the above CTRL-Binary maximin game (4). For simplicity and uniformity, here these will be formulated as equality constraints over rate reduction measures, but in practice they can be enforced softly during optimization.

**Sample-wise Self-Consistency via Closed-Loop Transcription.** First, to address the issue that CTRL-Binary does not learn a sample-wise consistent autoencoding, we need to promote $\hat{\mathbf{x}}$ to be close to $\mathbf{x}$ for each sample. In the CTRL framework, this can be achieved by enforcing that their corresponding features $\mathbf{z} = f(\mathbf{x})$ and $\hat{\mathbf{z}} = f(\hat{\mathbf{x}})$ are the same or close. To promote sample-wise self-consistency, where $\hat{\mathbf{x}} = g(f(\mathbf{x}))$ is close to $\mathbf{x}$ , we want the distance between $\mathbf{z}$ and $\hat{\mathbf{z}}$ to be zero or small, for all $N$ samples. This can be formulated using rate reduction; note that this again avoids measuring differences in the image space:

$$\sum_{i \in N} \Delta R(\mathbf{z}^i, \hat{\mathbf{z}}^i) = 0. \tag{5}$$

**Self-Supervision via Compressing Augmented Samples.** Since we do not know any class label information between samples in the unsupervised settings, the best we can do is to view every sample and its augmentations (say via translation, rotation, occlusion etc) as one "class" — a basic idea behind almost all self-supervised learning methods. In the rate reduction framework, it is natural to compress the features of each sample and its augmentations. In this work, we adopt the standard transformations in SimCLR (Chen et al., 2020b) and denote such a transformation as $\tau$. We denote each augmented sample $\mathbf{x}_a = \tau(\mathbf{x})$, and its corresponding feature as $\mathbf{z}_a = f(\mathbf{x}_a, \theta)$. For discriminative purposes, we hope the classifier is *invariant* to such transformations. Hence it is natural to enforce that the features $\mathbf{z}_a$ of all augmentations are the same as that $\mathbf{z}$ of the original sample $\mathbf{x}$. This is equivalent to requiring the distance between $\mathbf{z}$ and $\mathbf{z}_a$, measured in terms of rate reduction again, to be zero (or small) for all $N$ samples:

$$\sum_{i \in N} \Delta R(\mathbf{z}^i, \mathbf{z}_a^i) = 0. \tag{6}$$

### 3.3 Unsupervised Representation Learning via Closed-Loop Transcription

So far, we know the CTRL-Binary objective $\Delta R(\mathbf{Z}, \hat{\mathbf{Z}})$ in (4) helps align the distributions while sample-wise self-consistency (5) and sample-wise augmentation (6) help align and compress features associated with each sample. Besides consistency, we also want learned representations are maximally discriminative for different samples (here viewed as different "classes"). Notice that the rate distortion term $R(\mathbf{Z})$ measures the coding rate (hence volume) of all features. It has been observed in (Li et al., 2022) that by maximizing this term, learned features expand and hence become more discriminative.

**Unsupervised CTRL.** Putting these elements together, we propose to learn a representation via the following constrained maximin program, which we refer to as *unsupervised CTRL* (U-CTRL):

$$\max_{\theta} \min_{\eta} \quad R(\mathbf{Z}) + \Delta R(\mathbf{Z}, \hat{\mathbf{Z}}) \tag{7}$$

$$\text{subject to} \quad \sum_{i \in N} \Delta R(\mathbf{z}^i, \hat{\mathbf{z}}^i) = 0, \ \text{ and } \ \sum_{i \in N} \Delta R(\mathbf{z}^i, \mathbf{z}_a^i) = 0.$$

In practice, the above program can be optimized by alternating maximization and minimization between the encoder $f(\cdot, \theta)$ and the decoder $g(\cdot, \eta)$. We adopt the following optimization strategy that works well in practice, which is used for all subsequent experiments on real image datasets:

$$\max_{\theta} \ R(\mathbf{Z}) + \Delta R(\mathbf{Z}, \hat{\mathbf{Z}}) - \lambda_1 \sum_{i \in N} \Delta R(\mathbf{z}^i, \mathbf{z}_a^i) - \lambda_2 \sum_{i \in N} \Delta R(\mathbf{z}^i, \hat{\mathbf{z}}^i); \quad (8)$$

$$\min_{\eta} \ R(\mathbf{Z}) + \Delta R(\mathbf{Z}, \hat{\mathbf{Z}}) + \lambda_1 \sum_{i \in N} \Delta R(\mathbf{z}^i, \mathbf{z}_a^i) + \lambda_2 \sum_{i \in N} \Delta R(\mathbf{z}^i, \hat{\mathbf{z}}^i), \quad (9)$$

where the constraints $\sum_{i \in N} \Delta R(\mathbf{z}^i, \hat{\mathbf{z}}^i) = 0$ and $\sum_{i \in N} \Delta R(\mathbf{z}^i, \mathbf{z}_a^i) = 0$ in (7) have been converted (and relaxed) to Lagrangian terms with corresponding coefficients $\lambda_1$ and $\lambda_2$.[1]

**Unsupervised Conditional Image Generation via Rate Reduction.** The above representation is learned without class information. In order to facilitate discriminative or generative tasks, it must be highly structured. As we will see via experiments, specific and unique structure indeed emerges naturally in the representations learned using U-CTRL: globally, features of images in the same class tend to be clustered well together and separated from other classes (Figure 2); locally, features around individual samples exhibit approximately piecewise linear low-dimensional structures (Figure 5).

The highly-structured feature distribution also suggests that the learned representation can be very useful for generative purposes. For example, we can organize the sample features into meaningful clusters, and model them with low-dimensional (Gaussian) distributions or subspaces. By sampling from these compact models, we can conditionally regenerate meaningful samples from computed clusters. This is known as *unsupervised conditional image generation* (Hwang et al., 2021).

To cluster features, we exploit the fact that the rate reduction framework (1) is inspired by unsupervised clustering via compression (Ma et al., 2007), which provides a principled way to find the membership $\mathbf{\Pi}$. Concretely, we maximize the same rate reduction objective (1) over $\mathbf{\Pi}$, but fix the learned representation $\mathbf{Z}$ instead. We simply view the membership $\mathbf{\Pi}$ as a nonlinear function of the features $\mathbf{Z}$, say $h_{\boldsymbol{\pi}}(\cdot, \xi) : \mathbf{Z} \mapsto \mathbf{\Pi}$ with parameters $\xi$. In practice, we model this function with a simple neural network, such as an MLP head right after the output feature $\mathbf{z}$. To estimate a "pseudo" membership $\hat{\mathbf{\Pi}}$ of the samples, we solve the following optimization problem over $\mathbf{\Pi}$:

$$\hat{\mathbf{\Pi}} = \arg \max_{\xi} \Delta R(\mathbf{Z}|\mathbf{\Pi}(\xi)). \quad (10)$$

Experiments in Section 4.2 demonstrate that conditional image generation from clusters produced in this manner result in high-quality images that are highly similar in style.

## 4 EXPERIMENTS

We now evaluate the performance of the proposed U-CTRL framework and compare it with representative unsupervised generative and discriminative methods. The first set of experiments (Section 4.1 show that despite being a generative method in nature, U-CTRL can learn discriminative representations competitive with state-of-the-art discriminative methods. The second set (Section 4.2) show that the learned generative representation can significantly boost the performance of unsupervised conditional image generation. Finally, the third set (Section 4.3) study the advantages that generative represeentations have over discriminative ones.

We conduct experiments on the following datasets: CIFAR-10 (Krizhevsky et al., 2014), CIFAR-100 (Krizhevsky et al., 2009), and Tiny ImageNet (Deng et al., 2009). Standard augmentations for self-supervised learning are used across all datasets (Chen et al., 2020b).

We design all experiments to ensure that comparisons against U-CTRL are fair. For all methods that we compare against, we ensure that experiments are conducted with similar model sizes. If code for similar size structure can not be found, we uniformly use ResNet-18 to reproduce results for baselines, which is larger than the network used by our method. Details about network architectures and the experimental setting are given in Appendix A. All methods have runned 400 epochs or equivalent iterations (because generative models often count in iteration).

---

[1]Notice that computing the rate reduction terms $\Delta R$ for all samples or a batch of samples requires computing the expensive $\log \det$ of large matrices. In practice, from the geometric meaning of $\Delta R$ for two vectors, $\Delta R$ can be approximated with an $\ell^2$ norm or the cosine distance between two vectors.

| Method | CIFAR-10 | CIFAR-100 | Tiny-ImageNet |
|---|---|---|---|
| | Accuracy | Accuracy | Accuracy |
| *GAN based methods* | | | |
| SSGAN-LA(Hou et al., 2021) | 0.803 | 0.543 | 0.344 |
| DAGAN+(Antoniou et al., 2017) | 0.772 | 0.519 | 0.224 |
| ContraD(Jeong & Shin, 2021) | 0.852 | 0.514 | - |
| *VAE based methods* | | | |
| PATCH-VAE (Parmar et al., 2021) | 0.471 | 0.325 | - |
| $\beta$-VAE (Higgins et al., 2016) | 0.531 | 0.315 | - |
| *CTRL based methods* | | | |
| CTRL-Binary(Dai et al., 2022) | 0.599 | - | - |
| U-CTRL (ours) | **0.874** | **0.552** | **0.360** |

Table 2: Comparison of classification accuracy on CIFAR-10, CIFAR-100, and Tiny-ImageNet with other generative self-supervised learning methods. U-CTRL is clearly better.

| Method | CIFAR-10 | CIFAR-100 | Tiny-ImageNet |
|---|---|---|---|
| | Accuracy | Accuracy | Accuracy |
| SIMCLR | 0.869 | 0.545 | 0.359 |
| MoCoV2 | 0.872 | **0.589** | 0.365 |
| BYOL | **0.883** | 0.581 | **0.371** |
| U-CTRL (ours) | 0.874 | 0.552 | 0.360 |

Table 3: Comparison of classification accuracy on CIFAR-10, CIFAR-100, and Tiny-ImageNet with purely discriminative self-supervised learning methods. U-CTRL is close to these non-generative methods.

### 4.1 DISCRIMINATIVE QUALITY OF LEARNED REPRESENTATIONS

To evaluate the discriminative quality of the learned representations, we follow the standard practice of evaluating the accuracy of a simple linear classifier trained on the learned representation. Table 2 compares our method against SOTA generative self-supervised learning methods, and Table 3 compares our method against SOTA discriminative self-supervised methods. Experimental and training details are given in Appendix A.

**Quantitative Comparisons of Classification Performance.** From Table 2, we observe that on all chosen datasets, our method achieves substantial improvements compared to existing generative self-supervised learning methods. This includes more complex datasets like CIFAR-100 and Tiny-ImageNet, where we surpass the current SOTA models. From Table 3, our method achieves similar performance compared to SOTA discriminative self-supervised models. These results echo our goal of seeking a more unifed generative and discriminative representations: despite resembling a generative method architecturally, our method still produces highly discriminative representations. In addition, these results lead us to ask a fundamental question: when is incorporating both discriminative and generative properties greater than seperately handling these two parts? We provide preliminary answers in Section 4.3.

**Qualitative Visualization of Learned Representations.** To explain the classification performance of our method, we visualize the incoherence between features learned for the training datasets. Figure 2 shows cosine similarity heatmaps between the learned features, organized by ground-truth class labels. A block-diagonal pattern emerges automatically from U-CTRL training for all three datasets, similar to those observed in features learned in a supervised setting (Dai et al., 2022). In this case, however, these blocks emerge and correspond with classes labels despite the absence of any supervision at all.

### 4.2 IMPROVED UNSUPERVISED CONDITIONAL GENERATION QUALITY

To evaluate the quality of unsupervised conditional image generation, we measure performance on two axes: cluster quality and image quality. We estimate clusters by optimizing (10), and show results and comparisons with both recent and classical methods in Table 4. Training details of our method for the additional MLP head can be found in the Appendix A.

**Cluster Quality.** We measure normalized mutual information (NMI) and clustering accuracy for cluster quality on CIFAR-10 clustered into 10 classes and CIFAR-100(20), which is clustered into 20 super-classes. From Table 4, we observe that on CIFAR-10, U-CTRL results in an NMI that is almost

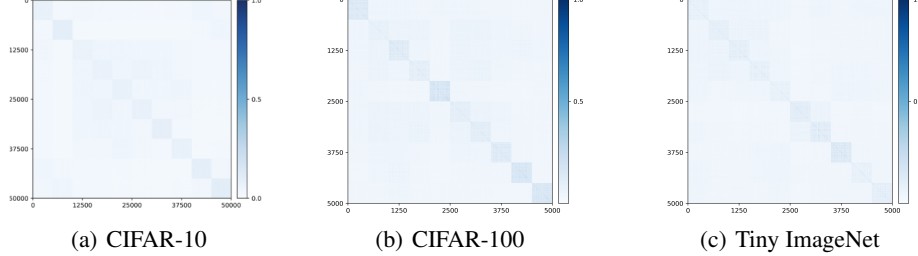

|    (a) CIFAR-10    |    (b) CIFAR-100    |    (c) Tiny ImageNet    |

Figure 2: Emergence of block-diagonal structures of $|\boldsymbol{Z}^\top \boldsymbol{Z}|$ in the feature space for CIFAR-10 (left), 10 random classes from CIFAR-100 (middle), and 10 random classes from Tiny ImageNet (right).

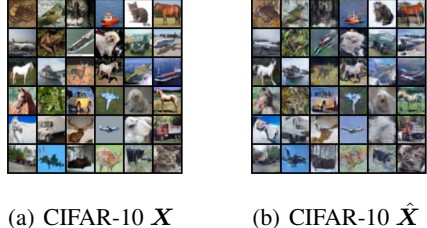

|    (a) CIFAR-10 $\boldsymbol{X}$    |    (b) CIFAR-10 $\hat{\boldsymbol{X}}$    |

Figure 3: Sample-wise self-consistency: visualization of images $\boldsymbol{X}$ and reconstructed $\hat{\boldsymbol{X}}$ on CIFAR-10 dataset.

double that of the existing SOTA on both GAN-based and VAE-based methods, with significantly improved clustering accuracy. Unlike many baselines, we also demonstrate that our method scales to the more challenging CIFAR-100(20) dataset, where it also significantly outperforms alternatives. Our improved clustering quality suggests potential for improving unsupervised conditional image generation, which relies on first finding statistically (and hence visually) meaningful clusters.

**Image Quality.** We use Frechet Inception Distance (FID) (Heusel et al., 2017) and Inception Score (IS) (Salimans et al., 2016) to measure image quality. From Table 4, it is evident that U-CTRL maintains competitive image quality compared to other methods, measured both by FID and IS. We also compare original images against reconstructed ones in Figure 3, where we see that the original $\boldsymbol{X}$ is very similar to the reconstructed $\hat{\boldsymbol{X}}$; U-CTRL indeed achieves very good sample-wise self-consistency.

**Unsupervised Conditional Image Generation.** In Figure 4, we visualize images generated from the ten unsupervised clusters from (10). Each block represents one cluster and each row represents one principal component for each cluster. Despite learning and training without labels, the model not only organizes samples into correct clusters, but is also able to preserve statistical diversities within each cluster/class. We can easily recover the diversity within each cluster by computing different principal components and then sample and generate accordingly! More detailed illustrations with more samples is provided in Appendix B.

### 4.3 BENEFITS OF U-CTRL'S STRUCTURED REPRESENTATION

As shown in the previous section, on datasets like CIFAR-10, CIFAR-100, and Tiny-ImageNet, our framework is able to achieve representation quality close to with the best discriminative self-supervised learning methods. A clear advantage of this is computational efficiency; only a single representation needs to be trained for a much broader set of tasks. This subsection aims to provide additional insights on how a unified model can be more beneficial for a broader range of tasks.

**Domain Transfer.** Regenerating images is demanding on the encoder, which is required to produce a more informative representation than contrastive training would. We hypothesize that the encoder trained with generative task may retain more information about the image and allow the representation to generalize better. To verify this, we compare the accuracy on CIFAR-100 using models learned from CIFAR-10 in Table 5. When compared to purely discriminative self-supervised learning models, we observe that U-CTRL is 4 percent better than other methods on classification accuracy.

**Visualization of Emerged Structures.** The representations learned by U-CTRL are significantly different from those learned from previous either discriminative and generative methods. To illustrate this, we use t-SNE (Van der Maaten & Hinton, 2008) to visualize the learned representation in 2D. Figure 5 compares the t-SNE of representations learned for CIFAR-10 by U-CTRL and MoCoV2,

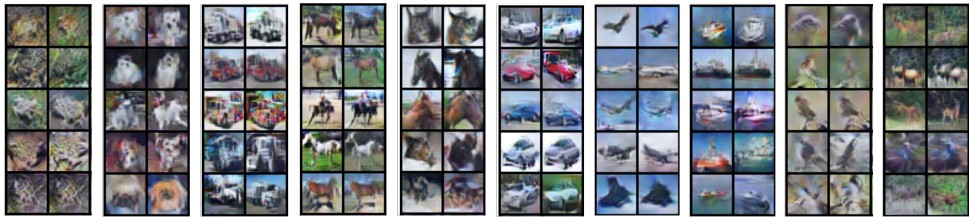

Figure 4: Unsupervised conditional image generation from each cluster of CIFAR-10, using U-CTRL. Images from different rows mean generation from different principal components of each cluster.

| Method | CIFAR-10 | | | | CIFAR-100(20) | | | |
|---|---|---|---|---|---|---|---|---|
| | NMI | Accuracy | FID↓ | IS↑ | NMI | Accuracy | FID↓ | IS↑ |
| *GAN based methods* | | | | | | | | |
| Self-Conditioned GAN (Liu et al., 2020) | 0.333 | 0.117 | 18.0 | 7.7 | 0.214 | 0.092 | 24.1 | 5.2 |
| SLOGAN (Hwang et al., 2021) | 0.340 | - | 20.6 | - | - | - | - | - |
| *VAE based methods* | | | | | | | | |
| GMVAE(Dilokthanakul et al., 2016) | - | 0.247 | - | - | - | - | - | - |
| Variational Clustering | - | 0.445 | - | - | - | - | - | - |
| *CTRL based methods* | | | | | | | | |
| U-CTRL (ours) | **0.658** | **0.799** | **17.4** | **8.1** | **0.374** | **0.433** | **20.1** | **7.7** |

Table 4: Comparison of the quality of UCIG on CIFAR-10 and CIFAR-100(20). Many of the methods compared do not provide code that scales up to CIFAR-100(20), in which case we leave the corresponding table cell blank.

respectively. It is clear that the representation learned by U-CTRL are highly structured and organized: classes are more evident, and features within each class form clear piecewise linear structures. We present more t-SNE comparisons in Appendix E.

| Method | SIMCLR | MoCoV2 | BYOL | U-CTRL |
|---|---|---|---|---|
| Accuracy | 0.422 | 0.436 | 0.437 | 0.481 |

Table 5: Comparing the transfer ability with purely discriminative self-supervised learning methods. All methods are trained unsupervised on CIFAR-10 and tested on CIFAR-100.

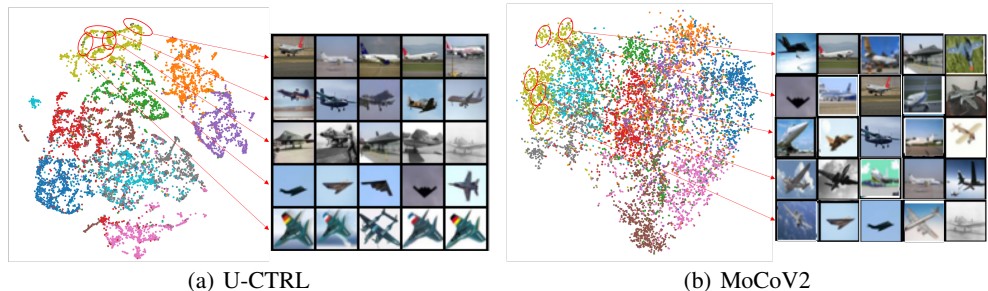

(a) U-CTRL      (b) MoCoV2

Figure 5: t-SNE visualizations of learned features of CIFAR-10 with different models.

## 5 CONCLUSION AND DISCUSSION

In this work, we proposed an unsupervised formulation of the closed-loop transcription framework (Dai et al., 2022). We experimentally demonstrate that it is possible to learn a unified representation for both discriminative and generative purposes, resulting in highly structured representations. Further, we show that these two purposes mutually benefit each other in various tasks, e.g., conditional image generation and domain tranfers. Compared to the more specialized representations learned in prior works, our results suggest that such a unified representation has the potential in supporting and benefiting a wider range of new tasks. In future work, we believe the learned representations can be further improved by jointly optimizing the feature representation and feature clusters, as suggested in the original rate reduction paper (Chan et al., 2022). Features with high likelihood of belonging to the same cluster can be further linearized and compressed. Due to its unifying nature and the simplicity of the underlying concepts, this new framework may be extended beyond image data, such as sequential or dynamical observations.

ETHICS STATEMENT

All authors agree and will adhere to the conference's Code of Ethics. We do not anticipate any potential ethics issues regarding the research conducted in this work.

REPRODUCIBILITY STATEMENT

Settings and implementation details of network architectures, optimization methods, and some common hyper-parameters are described in the Appendix A. We will also make our source code available upon request by the reviewers or the area chairs.

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

## A  Training Details

### A.1  Network Architectures

Table 6, 7 and Figure 6 give details on the network architecture for the decoder and the encoder networks used for experiments. The black rectangle marked with "conv, s=2" means a convlutional layer with stride 2. The orange rectangle marked with "dconv, s=2" means a deconvolutional layer with stride 2. The "x k" besides red frame means we regard these layers in red frame as a block and stack it k times. All $\alpha$ values in Leaky-ReLU (i.e. lReLU) of the encoder are set to 0.2. We set ($nz = 128$, $nc = 3$, $k = 3$) for CIFAR-10, ($nz = 256$, $nc = 3$, $k = 4$) for CIFAR-100, and ($nz = 256$, $nc = 3$, $k = 4$) for Tiny-ImageNet. As a comparison, ResNet-18 contains around 11 million parameters, whereas our encoder only contains between 4 and 6 million parameters depending on the choice of k.

Table 8 gives details of the network architecture for the linear classifier and Table 9 gives details of the network architecture for the additional MLP head used for unsupervised conditional image generation training.

| $z \in \mathbb{R}^{1 \times 1 \times nz}$ |
| --- |
| ResBlockUp. 256 |
| ResBlockUp. 128 |
| ResBlockUp. 64 |
| $4 \times 4$, stride=2, pad=1 deconv. 1 Tanh |

Table 6: Network architecture of the decoder $g(\cdot, \eta)$.

| Image $x \in \mathbb{R}^{32 \times 32 \times nc}$ |
| --- |
| ResBlockDown 64 |
| ResBlockDown 128 |
| ResBlockDown 256 |
| $4 \times 4$, stride=1, pad=0 conv $nz$ |

Table 7: Network architecture of the encoder $f(\cdot, \theta)$.

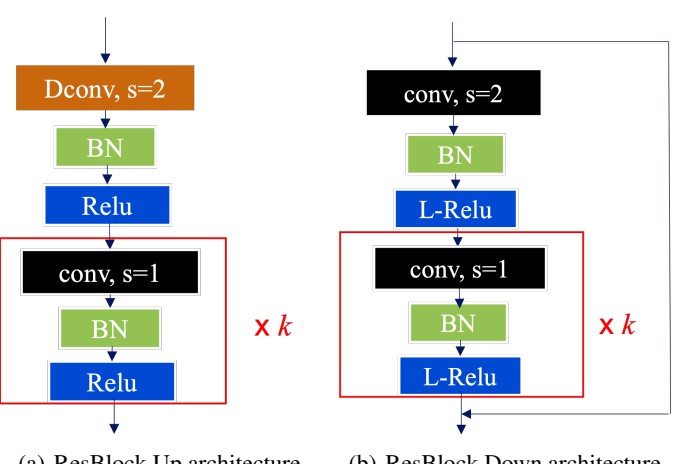

(a) ResBlock Up architecture      (b) ResBlock Down architecture

Figure 6: Architecture of two ResBlock.

### A.2  Optimization

For all experiments, we use Adam (Kingma & Ba, 2014) as our optimizer, with hyperparameters $\beta_1 = 0.5, \beta_2 = 0.999$. The learning rate is set to be 0.0001. We choose $\epsilon^2 = 0.2$. For all experiments, we adopt augmentation from SimCLR (Chen et al., 2020b).

| $z \in \mathbb{R}^{1 \times 1 \times nz}$ |
|:---:|
| Linear(nz, number of class) |

Table 8: Network architecture of the linear classifier.

| $z \in \mathbb{R}^{1 \times 1 \times nz}$ |
|:---:|
| Linear(nz, nz) ReLU |
| Linear(nz, number of clusters) |

Table 9: Network architecture of the MLP head for unsupervised conditional image generation

For CIFAR-10, CIFAR-100, and Tiny ImageNet, we train our framework with a batch size of 1024 over 20,000 iterations. All experiments are conducted with at most 4 RTX 3090 GPUs. Methods that are compared against in Table 3 are trained with the batch size of 256, because Chen et al. (2020b) observe that purely discriminative methods tend to perform better with smaller batch sizes. Table 2 methods have used their optimal parameters in their github code.

For training of the MLP head for unsupervised conditional image generation(10), we again use Adam (Kingma & Ba, 2014) as our optimizer with hyperparameters $\beta_1 = 0.5, \beta_2 = 0.999$. We choose the learning rate to be 0.0001 and $\epsilon^2$ as 0.2, with batch size 1024 over 5000 iterations.

For training of the linear classifier, we use Adam (Kingma & Ba, 2014) as our optimizer with hyperparameters $\beta_1 = 0.5, \beta_2 = 0.999$. We choose learning rate to be 0.0001, with batch size 1024 over 5000 iterations.

## B  ADDITIONAL UNSUPERVISED CLUSTERING AND GENERATION RESULTS

### B.1  CLUSTER RECONSTRUCTION

In this subsection, we visualize the reconstruction of ten clusters that are predicted and generated by U-CTRL on the CIFAR-10 training set. Each block in Figure 7 contains both a random sample of reconstructed data in a cluster and the total number of samples within it. Note that CIFAR-10 contains 50,000 training samples, split across 10 classes. As we see in Figure 7, the number of samples in each cluster are very close to 5,000, with the largest deviator (cluster 9) containing 3,942 samples. Without any cues, one can easily identify correspond each unsupervised cluster with a CIFAR-10 class. For a class like 'bird', we observe that the model is able to group images of standing birds, flying birds, and bird heads, despite their visual differences.

### B.2  UNSUPERVISED CONDITIONAL IMAGE GENERATION

Building on U-CTRL's ability to cluster CIFAR-10 samples, we demonstrate the model's ability to perform unsupervised conditional image generation in Figure 8. In contrast to reconstruction, where images are regenerated from features corresponding to real samples, we generate images based on the feature sampling technique proposed in (Dai et al., 2022). From these results, we observe that the U-CTRL framework maintains in-cluster diversity, and that the diversity can be recovered and visualized via simple principal component analysis.

## C  ABLATION STUDIES

### C.1  THE IMPORTANCE OF EACH TERM IN U-CTRL FORMULATION

In this section, we study the significance of the sample-wise constraints and extra rate distortion term in the formulation 7. Table 10 presents the following objectives that we study:

- Objective I is the constrained U-CTRL maximin 7.
- Objective II is the constrained maximin without the augmentation compression constraint 6.

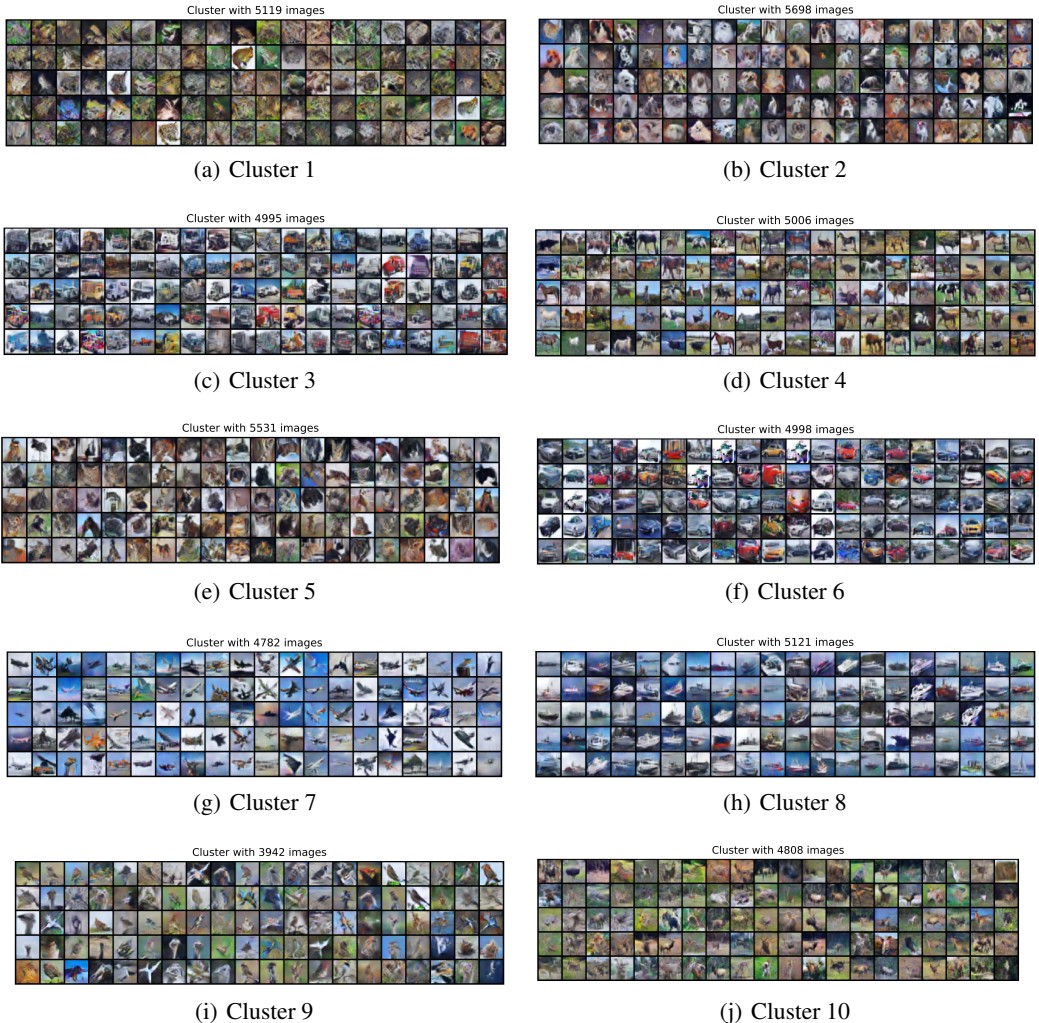

Figure 7: More result on the reconstruction of clusters in CIFAR-10

- Objective III is the constrained maximin without the sample-wise self-consistency constraint 5.
- Objective IV is the constrained maximin without the extra rate distortion term.
- Objective V is the U-CTRL without the augmentation compression constraint and sample-wise self-consistency constraint.
- Objective VI is the CTRL-Binary maximin formulation 4.

Table 11 shows the result of a linear probe for representations trained using each objective on CIFAR-10. From the table, it is evident that both constraints and the rate distortion term are pivotal to the success of our framework.

| | |
|---|---|
| Objective I: | $\max_\theta \min_\eta R(\boldsymbol{Z}) + \Delta R(\boldsymbol{Z}, \hat{\boldsymbol{Z}})$ s.t. $\sum_{i \in N} \Delta R(\boldsymbol{z}^i, \hat{\boldsymbol{z}}^i) = 0$, and $\sum_{i \in N} \Delta R(\boldsymbol{z}^i, \boldsymbol{z}_a^i) = 0$ |
| Objective II: | $\max_\theta \min_\eta R(\boldsymbol{Z}) + \Delta R(\boldsymbol{Z}, \hat{\boldsymbol{Z}})$ s.t. $\sum_{i \in N} \Delta R(\boldsymbol{z}^i, \hat{\boldsymbol{z}}^i) = 0$ |
| Objective III: | $\max_\theta \min_\eta R(\boldsymbol{Z}) + \Delta R(\boldsymbol{Z}, \hat{\boldsymbol{Z}})$ s.t. $\sum_{i \in N} \Delta R(\boldsymbol{z}^i, \boldsymbol{z}_a^i) = 0$ |
| Objective IV: | $\max_\theta \min_\eta \Delta R(\boldsymbol{Z}, \hat{\boldsymbol{Z}})$ s.t. $\sum_{i \in N} \Delta R(\boldsymbol{z}^i, \hat{\boldsymbol{z}}^i) = 0$, and $\sum_{i \in N} \Delta R(\boldsymbol{z}^i, \boldsymbol{z}_a^i) = 0$ |
| Objective V: | $\max_\theta \min_\eta R(\boldsymbol{Z}) + \Delta R(\boldsymbol{Z}, \hat{\boldsymbol{Z}})$ |
| Objective VI: | $\max_\theta \min_\eta \Delta R(\boldsymbol{Z}, \hat{\boldsymbol{Z}})$ |

Table 10: Five different objective functions for U-CTRL.

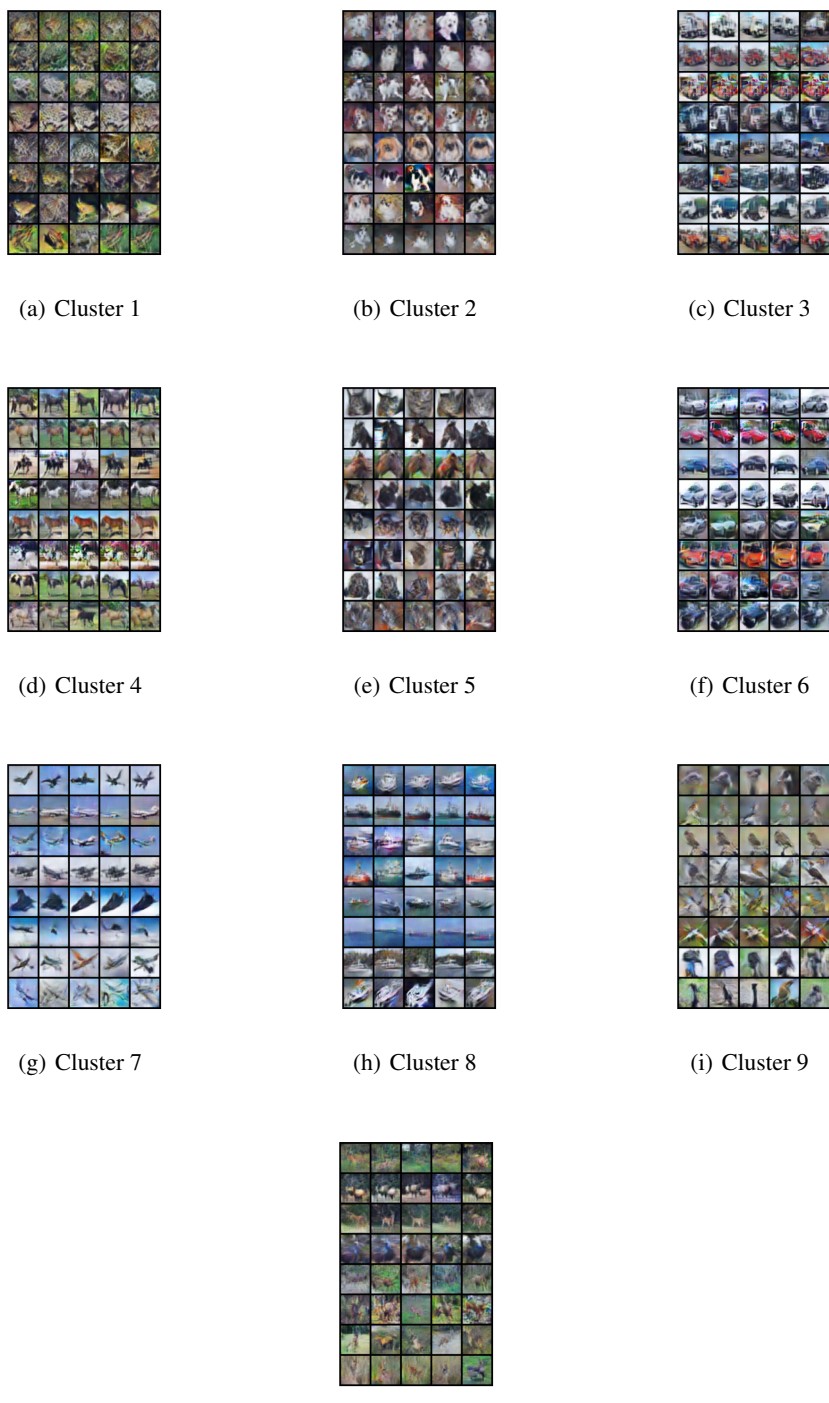

(a) Cluster 1          (b) Cluster 2          (c) Cluster 3

(d) Cluster 4          (e) Cluster 5          (f) Cluster 6

(g) Cluster 7          (h) Cluster 8          (i) Cluster 9

(j) Cluster 10

Figure 8: Unsupervised conditional image generation on CIFAR-10. Each block represents a cluster, within which each row represents one principal component direction in the cluster, and samples along each row represent different noises applied in that principal direction.

## C.2 THE IMPORTANCE OF MCR$^2$ IN U-CTRL FORMULATION

In this section, we verify the significance of MCR$^2$ term $\Delta R(\boldsymbol{Z}, \hat{\boldsymbol{Z}})$ in our method. We do ablation study on CIFAR-10 with the same network and training condition. If we take away MCR$^2$ from our

| Method | Objective I | Objective II | Objective III | Objective IV | Objective V | Objective VI |
|---|---|---|---|---|---|---|
| Accuracy | 0.874 | 0.578 | 0.644 | 0.522 | 0.633 | 0.599 |

Table 11: Ablation study on the significance of different terms in U-CTRL.

formulation, it changes (11). For simplicity, we call it U-CTRL-noMCR$^2$

$$\max_{\theta} \min_{\eta} \quad R(\boldsymbol{Z}) \tag{11}$$

$$\text{subject to} \quad \sum_{i \in N} \Delta R(\boldsymbol{z}^i, \hat{\boldsymbol{z}}^i) = 0, \quad \text{and} \quad \sum_{i \in N} \Delta R(\boldsymbol{z}^i, \boldsymbol{z}_a^i) = 0.$$

Table 12 shows that U-CTRL without the MCR$^2$ not only learns worse representation but also generalizes worse to out of distribution data. Figure 9 visualizes the reconstructed $\hat{\boldsymbol{X}}$ by U-CTRL-noMCR$^2$. It is clear from the image figure that without the MCR$^2$, the decoder fails to reconstruct high-quality images.

| | Accuracy on CIFAR-10 | Transfer Accuracy on CIFAR-100 |
|---|---|---|
| U-CTRL | 0.874 | 0.481 |
| U-CTRL-noMCR$^2$ | 0.836 | 0.418 |

Table 12: Ablation study on the significance of MCR$^2$ in U-CTRL.

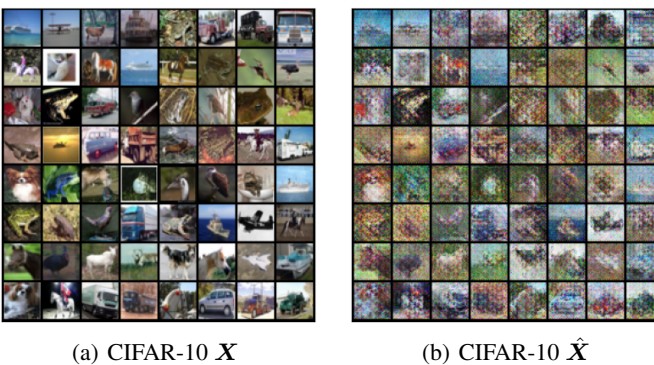

(a) CIFAR-10 $\boldsymbol{X}$        (b) CIFAR-10 $\hat{\boldsymbol{X}}$

Figure 9: Visualization of images trained by U-CTRL-noMCR$^2$: $\boldsymbol{X}$ and reconstructed $\hat{\boldsymbol{X}}$ on CIFAR-10 dataset.

It follows our discussion in the introduction that discriminative tasks and generative tasks together learn feature that benifits each other.

## D  RANDOM SEED SENSITIVITY

In this section, we verify the stability of our method against different random seeds. We report in Table 13 the accuracy of U-CTRL on CIFAR-10 with different seeds. We observe that the choice of seed has very little impact on performance.

| Random Seed | 1 | 5 | 10 | 15 | 100 |
|---|---|---|---|---|---|
| Accuracy | 0.874 | 0.876 | 0.870 | 0.874 | 0.871 |

Table 13: Ablation study on varying random seeds.

## E  MORE COMPARISON ON T-SNE

Due to limited space in the main body, we present a comparison of t-SNE between u-CTRL and other discriminative-based methods in this section. As shown in Figure 10, u-CTRL enjoys more structured representation comparing to purely discriminative methods.

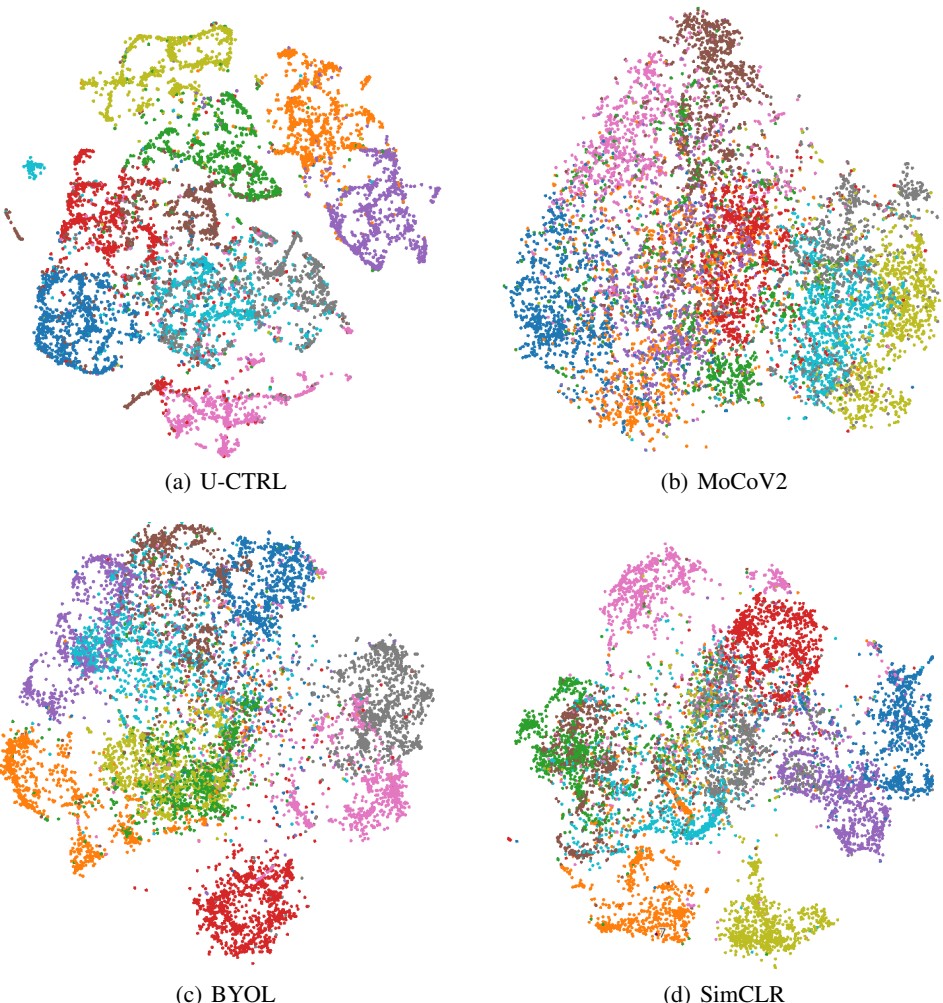

(a) U-CTRL

(b) MoCoV2

(c) BYOL

(d) SimCLR

Figure 10: t-SNE visualizations of learned features of CIFAR-10 with different models.

