# OpenReview forum: "Unsupervised Learning of Structured Representations via Closed-Loop Transcription"
_ICLR.cc/2023/Conference — Submitted to ICLR 2023_

### Official Review · Reviewer_TaSr · 2022-10-25

**Confidence:** 2
**Correctness:** 3
**Technical Novelty And Significance:** 3
**Empirical Novelty And Significance:** 3
**Recommendation:** 6

**Clarity, Quality, Novelty And Reproducibility:**


The paper is well written overall, and develops the approach well, and
I believe the results are reproducible.  There are several clarity
issues that I raised above.



**Strength And Weaknesses:**

Strengths:

-- A variety of experiments, on several tasks, show that the technique
   is superior than existing unsupervised generative or discriminative
   techniques.


Weaknesses:

-- The paper is dense, and I didn't come out with an understanding of
   why the method may work better.

-- The cost of optimization is not clear: how many rounds to
   convergence? (for 8 and 9)


Rewordings, etc:

-> (page 6, 'study how the..') Finally, the third set (Section 4.3)
study how the advantages that generative representations have over
discriminative ones.

-> (properties a whole greater) In addition, these results lead us to
 ask a fundamental question: when is incorporating both discriminative
 and generative properties a whole greater than the sum of its parts,
 particularly outside of the context of computational efficiency?

-> Why does superior clustering performance reflect the generative
   quality of the features: it seems to be that's another way of
   assessing the informativeness or the discriminative power of the
   features learned.

--> page 6: You mention 'structure' here and elsewhere: "In order to
facilitate discriminative or generative tasks, it must be highly
structured."  Do you just mean that the features come together to
reflect the underlying (hidden) class structure (rather than having structure
in the sense of having sub-parts?)



**Summary Of The Paper:**


The authors present an unsupervised approach, based on recent ideas
called 'closed-loop transcription', to extract feature encodings that
perform well on both discriminative and generative tasks, on CIFAR-10,
CIFAR-100 and Tiny Imagenet data sets.  Often there is a tradeoff
between generative vs discriminative goals, and their approach points
to the possibility of methods excelling at both (empirically).



**Summary Of The Review:**

This is a new idea, and shows good promise in a number of directions.
Many experiments and comparison are performed to showcase the
potential.

---

> ### Author Response · Authors · 2022-11-19
> **Response to Reviewer TaSr**
>
> We thank the reviewer for the constructive comments and suggestions! For the questions and suggestions, we have provided some clarifications below. Please let us know if you have any further questions or suggestions.
>
> >Q: Paper is dense, and I didn’t come out with an understanding of why the method work better:
>
>
> A: The existing CTRL framework works in the supervised learning setting and learns the  distribution of multiple classes jointly. Our work extends this closed-loop transcription framework to the much more general unsupervised setting by solving a constrained game. Compared to the representation learned by CTRL, U-CTRL can further enhance sample-consistency in the learned auto-encoding, hence resulting in significantly better conditional image generation, as Table 4 has shown. We have included some ablation studies on the significance of each term in our method.
>
> >Q: Cost of optimization is not clear.
>
> A: In optimization, 8 and 9 are optimized jointly, one after the other. We have updated the paper to be more clear on this point.
>
> > Rewording suggestions:
>
> A: Thank you for carefully going through our work and pointing out these suggestions! We have improved our paper based on your suggestions in the revised version.
>
>
> > You mention 'structure' here and elsewhere: "In order to facilitate discriminative or generative tasks, it must be highly structured." Do you just mean that the features come together to reflect the underlying (hidden) class structure (rather than having structure in the sense of having sub-parts?)
>
> A: In this work, we mean more than ”features come together to reflect the underlying class structure”. From Figure 4 and Figure.8, one can observe that our method has also learned clear low-dimensional structures \textbf{within} each class. We think it is a very important property that intuitively explains why the representation works well for both discriminative and generative purposes.
>
> > Why does superior clustering performance reflect the generative quality of the features:
>
> A: Unsupervised conditional image generation (UCIG) is measured in two parts: (1) to be able to find the correct clusters (2) generate high quality images. Therefore, superior clustering performance reflect model’s ability to perform UCIG. We reworded the main body of the paper to be more clear. We thank the reviewer for raising this point.
>
> We hope the above answers can clarify your questions. We would like to thank you again for the insightful questions and recognition of our work! Please let us know if you have any further questions

---

> ### Author Response · Authors · 2022-11-25
> **Further Discussion with the Reviewer**
>
> Dear Reviewer TaSr:
>
> We thank you for the precious review time and valuable comments. We have provided corresponding responses and results, which we believe have covered your concerns. We hope to further discuss with you whether or not your concerns have been addressed. Please let us know if you still have any unclear parts of our work.

---

### Official Review · Reviewer_jYoM · 2022-10-25

**Confidence:** 4
**Correctness:** 3
**Technical Novelty And Significance:** 3
**Empirical Novelty And Significance:** 3
**Recommendation:** 6

**Clarity, Quality, Novelty And Reproducibility:**

Clarity: This paper is clearly written (see strength 2).

Quality: Except for that this paper does not include ablation study (weakness 1), the empirical evaluation is fairly conducted and the results are convincing (strength 3).

Novelty: The novelty of this paper is from the auxiliary loss terms of $R(Z), \sum\limits_{i\in N} \Delta R(z^i, \hat{z^i}), \sum\limits_{i\in N} \Delta R(z^i, \hat{z}_a^i) $ (see weakness 1). This is another reason to conduct an ablation study on these terms.

Reproducibility: This paper provide all the experimental details to reproduce the results.

**Strength And Weaknesses:**

Strength:
1. The motivation is clear: one unified representation for both generative and discriminative purposes.
2. This paper is written clearly.
3. The empirical results are sufficiently strong enough to understand the effectiveness of the proposed method.

Weakness:
1. The proposed method is built upon the idea of CRTL-binary (i.e., $\max\limits_\theta \min\limits_\eta \Delta R(Z, \hat{Z})$).
To understand the contribution of this paper, ablation study of the additional loss terms (e.g., $R(Z), \sum\limits_{i\in N} \Delta R(z^i, \hat{z^i}), \sum\limits_{i\in N} \Delta R(z^i, \hat{z}_a^i) $  ) is highly recommended.
2. As noticed in the footnote of page 6, $ \Delta R(Z, \hat{Z})$ can be replaced with $\ell_2$ or cosine distance in practice and this trick cannot be applied to $R(Z)$ term. If rate reduction term is replaced with any simplified distances, it should be discussed in more details in the paper.

**Summary Of The Paper:**

- This paper illustrates a learning framework toward a unified representation which is effective for both generative and discriminative purposes.
- To do so, this paper introduce the idea of recent work, "Closed-Loop Transcription (CTRL) via a Constrained Maximin Game" in an unsupervised fashion.
- Specifically, the proposed method minimize the rate reduction between $Z$ and $\hat{Z}$, where $X \xrightarrow{f(x,\theta)} Z \xrightarrow{g(z,\eta)} \hat{X}  \xrightarrow{f(x,\theta)} \hat{Z}$, $f(x,\theta)$ is encoder and $g(z,\eta)$ is decoder.
- Then, the rate reduction between two code are defined as: $\Delta R(Z, \hat{Z}) \coloneqq R(Z \cup \hat{Z}) - \frac{1}{2}(R(Z) + R(\hat{Z}))$, where $R(Z)\coloneqq\frac{1}{2}\log\det(I+\frac{d}{N\epsilon^2}ZZ^\top) $.
- This paper also propose auxiliary loss term to 1) improve sample-wise self-consistency: $\sum\limits_{i\in N} \Delta R(z^i, \hat{z^i})$, and 2) self-supervision:  $\sum\limits_{i\in N} \Delta R(z^i, \hat{z}_a^i)$.
- Putting all this together, the unsupervised CTRL objective is defined as follows:
$\max\limits_\theta R(Z) + \Delta R(Z, \hat{Z}) - \lambda_1 \sum\limits_{i\in N} \Delta R(z^i, \hat{z^i}) - \lambda_2 \sum\limits_{i\in N} \Delta R(z^i, \hat{z}_a^i)$
- $\min\limits_\eta R(Z) + \Delta R(Z, \hat{Z}) + \lambda_1 \sum\limits_{i\in N} \Delta R(z^i, \hat{z^i}) + \lambda_2 \sum\limits_{i\in N} \Delta R(z^i, \hat{z}_a^i)$
- The experiments are conducted on three datasets: CIFAR10, CIFAR100, Tiny-ImageNet.
- The proposed U-CTRL shows clear performance gains over generative self-supervised learning methods.
- Its performance is also comparable to the discriminative self-supervised learning such as SimCLR, MOCO, BYOL.
- Its generative performance (UGIG task) is better than other generative models.
- By leveraging its generative objective, domain transfer performance is also better than other self-supervised methods.

**Summary Of The Review:**

This paper propose U-CTRL for learning an unified representation (generative and discriminative). The empirical results are very convincing, however, an ablation study is highly recommended to find their novelty over previous work.

---

> ### Author Response · Authors · 2022-11-19
> **Response to Reviewer jYoM**
>
> We thank the reviewer for the constructive comments and suggestions! For the questions and suggestions, we have provided some clarifications below. Please let us know if you have any further questions or suggestions.
>
> >Q1: Ablation Study on the additional loss term?
>
> A: We agree that it is crucial to justify the importance of additional loss terms with ablation studies.  Due to the limited space in the main paper. In the appendix Section C.1, we provide a very thorough ablation study on the importance of $R(Z)$, $\sum_{i\in N}\Delta R(z^i, \hat{z}^i)$, $\sum_{i\in N}\Delta R(z^i, z^i_a)$ and  as well as other pieces in our formulation. The result supports that the additional loss terms are important to the success of our method.
>
> >Q2: replace Delta R with l_2 or cosine similarity, discuss more:
>
> A: Thank you for raising this point! We have added more details about this point in the revised version of the paper.
>
> We hope the above answers can clarify your questions. We would like to thank you again for the insightful questions and recognition of our work! Please let us know if you have any further questions

---

> ### Author Response · Authors · 2022-11-25
> **Further Discussion with the Reviewer**
>
> Dear Reviewer jYoM
>
> We thank you for the precious review time and valuable comments. We have provided corresponding responses and results, which we believe have covered your concerns. We hope to further discuss with you whether or not your concerns have been addressed. Please let us know if you still have any unclear parts of our work.

---

### Official Review · Reviewer_Q2mk · 2022-10-26

**Confidence:** 4
**Correctness:** 3
**Technical Novelty And Significance:** 2
**Empirical Novelty And Significance:** 2
**Recommendation:** 5

**Clarity, Quality, Novelty And Reproducibility:**

- Clarity: Paper is clearly written
- Quality: Par with an average submission with promising early results.
- Novelty: Marginal at best, as the main contribution seems to be the two constraints described in eq 5 and eq 6., impact of which is not ablated in experiments.
- Reproducibility: Training details etc. are provided in the appendix, however the proposed method seems non-trivial in terms of training. Sharing code would be quite useful.

**Strength And Weaknesses:**

Strengths
- Strong performance against generative baselines, although on smaller datasets.

Weaknesses
- Unclear contribution: While the paper explicitly claims that it extends the CTRL framework to unsupervised setting, eq 4 already appears in Dai et al., 2022. Unless I am mistaken, CTRL framework as in Dai et al, 2022 is not limited to supervised setting only. This fact dilutes the claimed contribution. However, the incorporation of constraints (eq 5, 6) appears to be new.
- Limited Novelty: It seems that incorporation of self-consistency (eq 5) and augmentation (eq 6) are the main contributions of the paper. These on their own seem to be marginal at best. While experiments show that these do lead to superior results, some crucial ablations are missing (see next pt.)
- Missing ablations: Given that constraints listed in eq 5 and eq6 are the main contribution, these choices need to be studied. More specifically, how does the performance change from no constraint, to only first constraints, to only second constraint, to both of them?
- Inferiority to "discriminative" methods: The methods categorized under discriminative do not use labels (just like the proposed method), and appear to outperform it on the evaluated datasets. Further, more recent methods e.g. DINO [1], are superior to these baselines, while using a "simpler" training objective in comparison to the proposed method. This further reduces the appeal of the proposed method.
- Missing larger scale experiments: Most recent representation learning methods report their performance on the larger ImageNet dataset, to more closely mimic the practical setting. It would be beneficial to have that comparison to place the proposed work w.r.t. to these methods.



[1} Emerging Properties in Self-Supervised Vision Transformers, Caron et al.





**Summary Of The Paper:**

The paper claims to extend the previously proposed CTRL framework for representation learning to unsupervised setting. Paper proposes self-consistency and augmented sample compression based constraints to define an objective function for representation learning. Several experiments evaluate the representations learned by the proposed method by comparing against both discriminative and generative prior art.

**Summary Of The Review:**

I am leaning reject as the contributions seem marginal at best. Further, method is behind in terms of learned representation quality in comparison to discriminative methods. In addition, there are several other issues as detailed in the weaknesses section.

---

> ### Author Response · Authors · 2022-11-19
> **Response to Reviewer Q2mk**
>
> We thank the reviewer for the constructive comments and suggestions! For the questions and suggestions, we have provided some clarifications below. Please let us know if you have any further questions or suggestions.
>
> In general, we would like to point out that our method is *not* aiming towards producing a new SOTA performance. The main purpose, hence the novelty of this work, is seeking to learn a more unified representation, in an unsupervised setting, that can serve both discriminative and generative tasks and bridge the performance gap between learning discriminative and generative models separately.
>
> >Q: Unclear Contribution:
>
> A: The original CTRL framework (Dai et al, 2022) works in the supervised setting, whereas the contribution of this work extends the framework to the unsupervised or self-supervised setting. We show that the representation learned with the closed-loop framework, via a novel  constrained game formulation (contrary to an unconstrained game in Dai et al), serves well both classification and (conditional image) generation purposes. As one can observe, somewhat surprisingly,, we show that the so learned generative representation demonstrates classification performance close to discriminative methods (Table.2), better than state of the art generative methods (Table.3). For the task of  unsupervised conditional image generation, our method  also outperforms the current generative methods by a large margin (Table.4). We believe this is the first work to demonstrate such possibilities and such findings are valuable to the community. In the updated version of the main paragraph, we have re-written the contribution part to avoid misunderstandings.
>
> >Q: Inferior to “discriminative” methods, DINO for example:
>
> A: We would like to thank the reviewer for mentioning this very interesting new work. Nevertheless, we believe that DINO may not be an appropriate comparison in this case because our networks are based on traditional convolutional networks of small sizes, whereas DINO is based on vision transformers of much larger sizes. Due to the difference between sizes of networks and resources, it is not so fair to compare only the final performance of the two works. Nevertheless, we will discuss DINO in our final version.
>
> >Q: Missing ablation study on eq 5 and eq 6?
>
> A:  Due to limited space in the paragraph, we leave these ablation studies in the appendix. In section C.1, we provide a thorough ablation study on the importance of eq.5, eq.6.
>
> >Q: Missing large datasets
>
> A: We agree with the author that performance on ImageNet would closely mimic the practical setting more. However, due to the limited time of rebuttal period, we leave results on ImageNet to future studies. We think the results on the current choice of datasets is enough to verify the effectiveness of our method in pursuing a more unified representation for both discriminative and generative models. The choice of datasets are well justified following the common practice in published works like SLOGAN and SSGAN-LA.
>
> >Reference:
>
> [SLOGAN] Stein Latent Optimization for Generative Adversarial Networks, ICLR 2022
>
> [SSGAN-LA] Self-supervised gans with label augmentation, NIPS 2021
>
>
> We hope the above answers can clarify your questions. We would like to thank you again for the insightful questions! Please let us know if you have any further questions.

---

> > ### Comment · Reviewer_Q2mk · 2022-12-10
> > **Responses to the Author clarifications**
> >
> > Thank you for clarifying some of the details and I am adjusting my score based on those.
> >
> > > Contributions
> > Clarification from authors is helpful and I acknowledge that Dai et al. 2022 doesn't study unsupervised setting. I have incorporated that in my rating change.
> >
> > > Discriminative methods
> > While I mentioned DINO because it is presumably SOTA in this class, and counts as prior art w.r.t. this paper, there are earlier papers as well which used convolutional architectures e.g. [SwAV](https://arxiv.org/abs/2006.09882) from almost 2 years ago and are significantly better than the baselines used here (MOCO and SimCLR) in terms of reported results on Imagenet. Unfortunately, they don't report on CIFAR to make any direct comparison with current work feasible. I feel it is the responsibility of the authors to be aware of and compare with relevant prior art on relevant benchmark.  In a similar vein, at a high level I agree that comparing across architectures may not be fair, however, as the field advances it becomes necessary to keep up with recent advances for any proposed work to be relevant. In the current state with results on only small benchmarks and lack of comparison with these prior works, it is not possible for me to conclude that the results are indeed close to discriminative methods (as claimed in contributions)
> >
> > > Missing ablations
> > Thanks for the pointer. I have incorporated their existence in my rating change. However, I will note that for these to be really informative, we would ideally need to study different weightings as opposed to just existence of various terms.
> >
> > > Large datasets
> > Its unfortunate to not have this comparison on imagenet for representation learning as lots of prior art on this subject does report on that benchmark.

---

> ### Author Response · Authors · 2022-11-25
> **Further Discussion with Reviewer**
>
> Dear Reviewer Q2mk:
>
> We thank you for the precious review time and valuable comments. We have provided corresponding responses and results, which we believe have covered your concerns. We hope to further discuss with you whether or not your concerns have been addressed. Please let us know if you still have any unclear parts of our work.

---

### Official Review · Reviewer_K48e · 2022-11-01

**Confidence:** 4
**Correctness:** 3
**Technical Novelty And Significance:** 2
**Empirical Novelty And Significance:** 2
**Recommendation:** 5

**Clarity, Quality, Novelty And Reproducibility:**

In terms of clarity, I really enjoyed reading the manuscript. Most parts of this manuscript is clearly well-written. And, in terms of reproducibility, many details are well described, so I believe that all the results could be reproducible.

In terms of quality and novelty, as described in the section above, I’ve raised some issues on the experiments and motivation.

**Strength And Weaknesses:**

**Strengths**:

* This manuscript is basically very well-written, and clearly summarizes the necessary previous works to understand the main argument. It's easy to follow most of the parts in the manuscript.
* It seems that the approach based on closed-loop transcription and rate-distortion theory for learning better representation is quite principal.
* This manuscript includes all the details in the experiments, including detailed setup, hyperparameters, and network configuration. It seems that all the results are reproducible.


**Weaknesses**:

My biggest concern on this work is the low performance of discriminative learning shown in Table 2. Compared to the similar baselines, Table 2 is not enough to argue that the learned representation by U-CTRL is highly discriminative. I extracted the linear evaluation experiments in GCRL and W-MSE, as shown in below:


|Method| CIFAR-10 | CIFAR-100 | Tiny ImageNet|
|----------|----------|----------|----------|
|U-CTRL| 87.4 | 55.2 | 36.0 |
|GCRL | 83.9-95.1 | 58.7 - 76.0 | - |
|W-MSE | 91.6 - 92.0 | 66.1 - 67.6 | 48.2 - 49.2 |

Besides the comparison to GCRL and W-MSE, I’ve found that the performance of BYOL in Table 2 is much worse than the reported numbers in Table 1 in W-MSE paper. Could you explain a little bit more why the performance gap is observed?

I’m also still not convinced that the learned representation based on U-CTRL is more structured than SSL methods, especially the methods based on contrastive losses. Therefore, it could be necessary to measure the cluster quality of representation learned by contrastive losses, since the contrastive loss enforces the semantically-meaningful clusters in the representation space. In this work, U-CTRL was only compared to the learned representation based on generative learning, which is not sufficient to argue that the representation from U-CTRL is more structured than the ones from existing approaches. Figure 5 seems to support the argument, but it could be better to include the results of SimCLR and BYOL.

This work also argues that the structured representation improves the quality of generated samples a lot. However, as shown in unCLIP (a.k.a. DALL-E 2), the representation based on a simple contrastive loss definitely helps to train a decoder. It means that the learned representation based on contrastive losses is already structured enough to train a generative model.

References:
* [W-MSE] Whitening for Self-Supervised Representation Learning, ICML’21.
* [GCRL] Hybrid Generative-Contrastive Representation LEarning, arXiv’21.
* [unCLIP] Hierarchical Text-Conditional Image Generation with CLIP Latents, arXiv’22.

**Summary Of The Paper:**

This work proposes an unsupervised extension of the closed-loop transcription, named as U-CTRL, to learn the better representation for discriminative and generative learning tasks. The authors show that the learned representation is more structured, i.e. having the cluster shapes in semantically-meaningful way, improving the performance of linear evaluation and conditional generative tasks.

**Summary Of The Review:**

My initial evaluation is borderline, slightly leaning towards rejection. My major concern on this work is the insufficient empirical justification. In specific, I’m not sure that the learned representation based on U-CTRL is discriminative enough compared to the similar recent works. In addition, I failed to figure out that the representation learned by U-CTRL is more structured than the ones by contrastive losses.

---

> ### Author Response · Authors · 2022-11-19
> **Response to Reviewer K48e**
>
> We thank the reviewer for the valuable comments and suggestions! For the comment you have, we have provided some clarification and additional experiments to answer them. We have also updated the paper based on your comments.
>
> >Q1:  Performance gap between the performance on Table.2 in our method and Table.1 in W-MSE work?
>
> A: Thank you for raising this concern. In our work, Table 2 reports results of the methods trained for 400 epochs, whereas Table 1 in W-MSE contains results trained for 1000 epochs. We adopt this more commonly used setting for comparison as in DCL. Note that our method may not be the best for discriminative tasks, especially when compared with purely discriminative models. As shown in Table.2, we have the highest linear probing accuracy compared within generative based SSL methods. The value of our work is precisely to show that a common representation learned in an unsupervised way can largely bridge the performance  gap between representations learned for discriminative or generative tasks only.
>
>
> >Q2: Measure cluster quality of other work(Experimenting):
>
> A: Following your suggestion, we tried to use equation 10 to measure the cluster quality of representation learned by contrastive loss. However, we discovered that our cluster method is not applicable to these contrastive-discriminative methods. For fair comparison, we adopt semantic clustering algorithms from SCAN (the SOTA clustering algorithm) to measure the cluster quality of contrastive discriminative methods.
>
> |                     | SimCLR | MoCoV2 | BYOL  | u-CTRL(ours) |
> |---------------------|--------|--------|-------|--------------|
> | Clustering Accuracy | 0.787  | 0.790  | 0.806 | 0.799        |
>
> Similar to the results on linear probing, the representation learned by our method has comparable clustering performance to contrastive-discriminative methods.
>
> >Q3: Comparing t-SNE plot with other methods such as BYOL and SimCLR?
>
> A:Following your suggestion, we have added a section in Appendix.E comparing t-SNE visualization of our method with BYOL, SimCLR and MOCOV2. From t-SNE visualization, we observe that U-CTRL learns more refined inter-class and intra-class structures. Nevertheless, we agree that t-SNE alone might not be conclusive, so we have also changed the corresponding text in the main body to claim only that the representation learned by U-CTRL is \textbf{highly} structured. We believe an unsupervised generative model with a highly structured representation is a valuable result to the generative model community.
>
> >Q4: Learned representation based on contrastive loss is already structured enough to train a generative model like DALL-E 2
>
>
> A: This is a very interesting point. Yes, it seems that methods like DALL-E 2 learn “structured” representations from contrastive loss. Nevertheless, such structures are acquired with additional costs: First, such methods require images data paired with text. Hence the “structured’’ representations are learned with “soft supervision” by the text labels, whereas our method is entirely unsupervised.  In other words, our model learns explicit structure in the latent representation whereas DALL-2 learns implicit structure via text-conditioning. Second, the amount of resources, data, network sizes required by contrastive methods such as DALL-E 2 are several magnitudes more than what is required by ours. In addition, DALL-E 2 is a complex system with multiple encoders and decoders trained for both vision and language data. In comparison, our method uses a simple unified closed-loop framework, and learns a representation in a purely unsupervised way that serves both discriminative and generative purposes.  To our best knowledge, representations learned by DALL-E cannot be used for classification.
>
> We hope the above answers can clarify your questions. We would like to thank you again for the insightful questions! Please let us know if you have any further questions.
>
> References:
> [DCL] Decoupled Contrastive Learning, ECCV 2022, Chun-Hsiao Yeh, Cheng-Yao Hong, Yen-Chi Hsu, Tyng-Luh Liu, Yubei Chen, Yann LeCun

---

> ### Author Response · Authors · 2022-11-25
> **Further Discussion with the Reviewer**
>
> Dear reviewer K48e:
>
> We thank you for the precious review time and valuable comments. We have provided corresponding responses and results, which we believe have covered your concerns. We hope to further discuss with you whether or not your concerns have been addressed. Please let us know if you still have any unclear parts of our work.

---

> ### Author Response · Authors · 2022-12-10
> **Thank you for reviewing our work**
>
> Dear Reviewer K48e,
> Since the discussion phase is about to end in a few days, we wonder if our reply properly addressed the concerns. Thank you for your valuable time and helpful suggestion!

---

### Decision · Program_Chairs · 2023-01-20

**Decision:**

Reject

**Justification For Why Not Higher Score:**

The authors did a good job of updating this paper, but it is still a bit borderline, and I think that some of the additional comparisons and discussions requested by the reviewers are fair.

However, I would have no objection to seeing this accepted.

**Justification For Why Not Lower Score:**

N/A

**Metareview: Summary, Strengths And Weaknesses:**

This paper introduces a new approach for unsupervised feature learning, which is motivated by the "closed-loop transcription" framework, CTRL. Overall it is quite different from many other current popular self-supervised or autoencoder-based methods for feature pretraining, and as such I think it is liable to be of general interest. Concerns from the reviewers focused on novelty (relative to the original supervised CTRL paper), and on the importance of the empirical results (particularly a lack of ablations of the two Lagrangian terms, other potential baselines, and worse-than-expected discriminative performance). The author response went some ways towards addressing these concerns; at least one reviewer raised their score. However, the overall set of recommendations is still mixed.

**Summary Of Ac-Reviewer Meeting:**

This was not flagged as a borderline paper, as at the time papers were selected for discussion the average score was quite a bit lower (reviewers have since raised their scores).